# Prevalence of Perceived Stress, Anxiety, and Depression in HCW in Kosovo during the COVID-19 Pandemic: A Cross-Sectional Survey

**DOI:** 10.3390/ijerph192416667

**Published:** 2022-12-12

**Authors:** Fatime Arenliu Qosaj, Stevan Merrill Weine, Pleurat Sejdiu, Fekrije Hasani, Shukrije Statovci, Vigan Behluli, Aliriza Arenliu

**Affiliations:** 1Master Program in Healthcare Management, College AAB, 10000 Prishtina, Kosovo; 2Center for Global Health, University of Illinois, Chicago, IL 60612, USA; 3Kosovo Medical Chamber, 10000 Prishtina, Kosovo; 4Orthopedic Clinic, Kosovo Hospital University Clinical Services, 10000 Prishtina, Kosovo; 5Alma Mater Europaea, Campus College Rezonanca, 10000 Prishtina, Kosovo; 6Kosovo Nursing Chamber, 10000 Prishtina, Kosovo; 7Ministry of Health, Government of the Republic of Kosovo, 10000 Prishtina, Kosovo; 8Psychiatry Clinic, Kosovo Hospital University Clinical Services, 10000 Prishtina, Kosovo; 9Kosovo Association of Psychologists, 10000 Prishtina, Kosovo; 10Department of Psychology, Prishtina University, 10000 Prishtina, Kosovo

**Keywords:** health care worker, COVID-19, pandemic, mental health, depression, anxiety, stress, risk factors, protective factors

## Abstract

A pandemic may have a negative impact on healthcare workers’ (HCW) mental health. In this cross-sectional study, we assess the self-reported prevalence of stress, anxiety, and depression and identify their predictive factors among HCW in Kosovo. The online questionnaire collected data on socio-demographics (sex, age, occupation, education, workplace) and the presence and severity of depression, anxiety, and stress through the 21-item Depression, Anxiety, and Stress Scale (DASS-21) questionnaire. Descriptive statistics, t-test, and linear logistic regression were used to analyze the data. Of the 545 respondents, the majority were male (53.0%), under 60 years of age (94.7%), and married (81.7%). Most of them were physicians (78.2%), while the remaining were nurses, midwives, and other health professionals (22%). Prevalence rates for moderate to extremely high stress, anxiety, and depressive symptoms were 21.9%, 13.0%, and 13.9%, respectively. The nurses reported significantly higher mean scores for depression and anxiety than the physicians (*p* < 0.05). Being married, having poor health, not exercising, and reporting “burnout” from work significantly predicted higher levels of depressive, anxiety, and stress symptoms among health workers (*p* < 0.05). Most HCWs (71.6%) reported a mild, moderate, or severe mental health burden, and certain factors predicted higher levels of such burden.

## 1. Introduction

The COVID-19 pandemic has created the anticipation of a substantial workload burden on health systems worldwide. COVID-19 cases worldwide are continuing to challenge global- and country-level strategies to help control, end, and prepare for the next possible pandemic. Kosovo is no exception to this trend.

In 2019, at the global level, of 369 diseases in 204 countries, mental health disorders (depressive disorders, self-harm, and anxiety) were ranked among the 25 most common causes of Disability-adjusted life years (DALYs). These mental health disorders caused 4.2% of the DALYs and were the 4th most common cause after neonatal deaths, ischemic heart disease, and stroke [1]. On the other hand, the emergence of the COVID-19 pandemic at the global level has increased the burden of major depressive disorder and anxiety symptoms by 27.6% and 25.6%, respectively [2]. On the other hand, it is known that long periods of exposure to adverse determinants of mental health may lead to the cumulative effects of stressful experiences in daily life (allostatic load), which may further lead over time to diseases, i.e., cardiovascular diseases, hyperglycemia, fibromyalgia, and breast cancer [3]. Therefore, there is an urgent increase in the necessity to support the broader provision of mental health services and target determinants of poor health and mental health disorders.

Health care workers (HCWs) have been at the forefront of all pandemics. Many studies have reported increased psychological problems among HCWs during and after earlier H1N1, SARS, and Ebola epidemics [4,5,6,7,8,9]. To this end, during the ongoing COVID-19 pandemic, many global [10], multinational [11,12,13,14], and country-level studies [15,16,17,18,19,20,21,22,23,24] continued to document the high prevalence of self-reported depressive, anxiety, and stress symptoms among HCW. Moreover, some emerging studies have tried to understand the main adverse determinants of psychological outcomes related to working contexts, such as intense long working hours, working with infected patients [25], concerns about having insufficient adequate personal protective equipment (PPE) in stock [26,27], and fear of the risk of being infected and transmitting the virus to the members of their family and the community where they live and work [25]. An international study in 41 countries indicated that redeployment of HCWs to the intensive care unit with unsatisfactory training and poor workplace support was significantly negatively associated with mental health symptoms during the COVID-19 pandemic [27]. Burnout due to workload, i.e., a feeling of energy depletion or exhaustion, increased mental distance from one’s job, or feelings of negativism or cynicism related to one’s career, and reduced professional efficacy among HCWs were reported during previous [28,29] and COVID-19 [28,30,31,32,33,34] pandemics.

Among the countries affected by the COVID-19 pandemic in the Balkans, Kosovo was the last country to report its first COVID-19 case on 13 March 2020. Since then, Kosovo has experienced five waves of the pandemic [35]. During this time, authorities were challenged between containing the spread of the virus and mitigating the adverse effects of restrictions on the economy and social life. Since March 2020, following the COVID-19 trends, lockdown measures have been applied accordingly. At the beginning of the pandemic, the Kosovo government adopted lockdown measures from March to May 2020. Measures gradually phased out from 4 May to 1 July 2020, when almost all businesses were allowed to reopen [36]. These measures continued to require the maintenance of hygienic standards in public and private institutions, wearing masks, and respecting physical and social distancing [37]. No studies report the prevalence of burnout due to workload among HCWs in Kosovo. One study in Kosovo among HCW found a very high percentage of HCWs with severe levels of anxiety (44.6%) and depressive symptoms (38.7%) [38].

This study attempts to understand the mental health symptoms of HCWs during the COVID-19 pandemic and to provide evidence for the mental health interventions required. The study’s specific aims were to explore the levels of and factors associated with self-reported perceived stress, anxiety, and depression HCW during the early phase of the COVID-19 pandemic.

## 2. Materials and Methods

### 2.1. Ethics

Participant consent was indicated via submission of the HCW survey responses. It was clearly stated that participation in the study was voluntary, with the possibility of opting out at any time or choosing not to answer specific questions, with no negative consequences. Ethical approval for the research was obtained through the Research Ethics Committee of the Kosovo Medical Chamber (Ref. no. 12/2020).

### 2.2. Study Design and Sampling

A cross-sectional, descriptive, and correlational survey was conducted to explore the self-reported prevalence of perceived stress, anxiety, depressive symptoms, and their associated protective and risk factors in HCW (19,974) registered with the Kosovo Medical and Nursing Chamber. The questionnaire was submitted through the Kosovo Medical and Nursing Association. The sampling period (30 September to 30 October 2020) was at its lowest point one week before the third wave peak of the pandemic in Kosovo.

### 2.3. Measurement Tools

The survey questionnaire was sent to the registered HCW through the Medical and Nursing Chambers mailing lists. The e-mails that were sent contained an explanatory and consent message, with a link to a 15–20-min online survey requesting demographic (age, gender, marital status), work (education, profession, burnout, availability of PPE), health (self-reported BMI, self-rated health, and self-reported chronic diseases, COVID-infection status and physical exercise) data and clinical characteristics using validated tools (self-reported stress, anxiety, depression).

Clinical characteristics were assessed using validated screening scales for self-reported symptoms of stress, anxiety, and depression (DASS-21; DASS-21 scores ≥ 14–20, ≥21–27, and ≥28–42 indicate moderate, severe, and extremely severe stress, anxiety, and depression, respectively) [39]. Validated scales were used to understand HCW self-reported symptoms and potential severity and screen for symptomatology. These scales were not intended as diagnostic tools.

### 2.4. Data Analyses

Data analysis was undertaken using SPSS Statistics for Windows (Version 26; IBM Corp) [40]. The demographic characteristics of the respondents and the prevalence of self-perceived stress, anxiety, and depression mean absolute numbers and percentages summarized scores. We used *t*-test analyses with two-tailed significance levels (*p* < 0.05) to assess for significant differences in mean scores for self-perceived stress, anxiety, and depression.

Correlation analyses were performed before the logistic linear regression analysis to rule out powerful correlations among predictor variables. Odds ratios from the binary logistic regression analyses were examined to determine the associations between demographic, work, and physical health factors, and the likelihood of respondents self-reporting symptoms of stress, anxiety, and depression. There were no imputations for missing data, and the results presented reflect completed responses from the survey.

## 3. Results

### 3.1. Participants’ Characteristics

Overall, 19,974 surveys were delivered through the internet to the registered members of Kosovo Medical and Nurse Chambers (100%). Of the 545 respondents, a majority were male (*n* = 289, 53.0%), under 60 years of age (*n* = 508, 94.7%), and married (*n* = 445, 81.7%). The majority were medical doctors (*n* = 426, 78.2%), while the remaining respondents were nurses, midwives, and other health professionals (*n* = 118, 22%), who mainly worked in hospitals (*n* = 144, 26.4%), clinics (*n* = 141, 25.9%), and primary health care facilities (184, 33.8%). Most had 11 or more years of working experience (*n* = 363, 66.5%).

The 4-week prevalence rates in the sample (*n* = 545) for stress, anxiety, and depression were 21.9%, 13.0%, and 13.9%, respectively. The descriptive demographic characteristics of the survey respondents are displayed in Table 1. There was no imputation for missing data, and the data analyzed and displayed were from the dataset that remained after cases with missing data were excluded.

### 3.2. Prevalence of Stress, Anxiety, and Depression

Table 2 shows that HCW reported normal (0–9), mild (10–13), moderate (14–20), and extremely severe (21–27) levels of self-reported prevalence of stress, anxiety, and depression.

As shown in Table 3, we assessed the difference in mean scores for self-perceived stress, anxiety, and depressive symptoms for nurses, other health care professionals, and doctors. Nurses reported significantly higher average mean scores for depression (M = 5.16, SD = 4.64) and anxiety (M = 4.85, SD = 4.79) than doctors (M = 4.19, SD = 4.65, t (542) = 2.00, *p* = 0.04; M = 3.86, SD = 4.23, t (542) = 2.04, *p* = 0.04, respectively). No significant difference (t (542) = −4.56, *p* = 0.65) was found in average scores for stress, where nurses scored (M = 5.80, SD = 4.56) and medical doctors scored (M = 6.02, SD = 4.94).

### 3.3. Correlational Analysis

Table 4 displays descriptive statistics and correlation coefficients between the relevant observed variables. The self-reported depression, anxiety, and stress DAS-21 scores were negatively associated with physical activity, chronic disease, and availability of PPE. The DASS-21 depression scores were negatively associated with physical exercise (r = −0.162, *p* < 0.01), self-reported chronic disease (r = −0.216, *p* < 0.01), and availability of PPE (r = −0.135, *p* < 0.01). However, depression scores were positively associated with burnout (r = 0.248, *p* < 0.01) and self-reported BMI (r = 0.109, *p* < 0.05).

The DASS-21 anxiety scores were negatively associated with physical exercise (r = −0.142, *p* < 0.01), self-reported chronic disease (r = −0.197, *p* < 0.01), and the availability of PPE (r= −0.146, *p* < 0.01). However, anxiety scores were positively associated with burnout (r = 0.213, *p* < 0.01).

The DASS-21 stress scores were negatively associated with physical exercise (r = − 0.148, *p* < 0.01), chronic disease (r = −0.129, *p* < 0.01), and availability of PPE (r = −0.139, *p* < 0.01). However, stress scores were positively associated with burnout (r = 0.318, *p* < 0.01).

### 3.4. Linear Modeling

To assess the impact of predictors on self-reported stress, anxiety, and depression scores while controlling for demographic characteristics, we entered all characteristics in Table 4 and those related to COVID-19 infection status into the logistic model.

Table 5 shows the impact of sociodemographic, work, and health-related factors on the self-reported depression, anxiety, and stress scale scores among the HCW. All three linear modeling analyses showed that being married, working in a hospital, having an adequate continuous supply of PPE, feeling workplace burnout, the level of self-rated health, and engaging in regular physical exercise were statistically significant factors explaining the self-reported stress, anxiety, and depression scores. Having a family member with COVID-19 contributed substantially to explaining the self-reported stress score but not the self-reported depression and anxiety scores. While being a nurse was considered statistically significant in explaining the self-reported anxiety and depression, but not the self-reported stress. In all three linear modeling analyses, gender, offering services to COVID-19 patients, and being infected with SARS-CoV-2 were not statistically significant in explaining self-reported depression, anxiety, and stress scores.

## 4. Discussion

Our study explored the prevalence of perceived stress, anxiety, and depression in HCW in Kosovo during the COVID-19 pandemic. We found that most HCWs (71.6%) reported a mild, moderate, or severe mental health burden. However, most doctors, nurses, and other HCWs reported a normal to mild level of DASS, which concurred with previous studies [13,38,41]. The distress levels among the Kosovo HCWs were comparable with those in Italy [41] and were higher than those reported in Singapore [11]. We reported lower levels of depression and anxiety than the initial findings obtained in Kosovo [38]. This difference may be explained by different screening tools [42] and different samples. Moreover, the period between April and October 2020, when the two surveys were conducted, corresponds to the first-wave peak and second-wave pre-peak of the COVID-19 pandemic in Kosovo. This period ranged from strict lockdown measures (March–April 2020) to relaxing phased-out measures (May–July 2020) [36]. However, compared to the studies conducted in Europe using the same survey tool, our findings were within the range of reported DASS-21 scales [13]. The lower levels of psychological distress reported in October [38] as compared to those found in our study among HCWs in Kosovo could also be attributed to the decreasing COVID-19 epidemic peak, strain at work, and Kosovo’s relatively low number of new cases per one million people [43] compared to other studied countries [13]. The prevalence of depression, anxiety, and stress symptoms in our study was lower than that in multinational-level studies [12,13,14]. The prevalence of DAS symptoms found in Kosovo falls within the prevalence ranges reported for countries at the global level [10]. Namely, the prevalence of depression found in Kosovo falls within the lowest levels of depression and stress reported for countries at the global level and can be compared with stress levels reported in South Korea and Iran [10].

Our analyses indicated that being married, being a nurse, working in a hospital, continuously lacking adequate PPE, and feeling workplace burnout significantly predicted depression, anxiety, and stress scores. These findings are in line with contextual realities in Kosovo. Escalating workloads during the COVID-19 pandemic [43], chronic low health spending [44], and weak managerial capacities to manage efficiently and motivate HCW [45] may have further increased the challenge of providing needed care to patients, resulting in HCW burnout. These systematic shortfalls have increased the chance of migrating after the pandemic, especially among nurses, other healthcare workers, and those working at secondary and tertiary levels [46]. Subsequently, HCWs migration may further challenge the replacement of HCWs that will be retiring in forthcoming years. Our findings are in agreement with a meta-analysis on the correlates of depressive symptoms in more than 14,000 HCWs amid the first phases of the COVID-19 pandemic. Namely, nurses feeling not adequately protected were more likely to report depression as compared to doctors [47].

Other variables, having COVID-19, having a member of the family with COVID-19, and offering services to COVID-19 patients, albeit considered important [47], were not found to be significant factors in explaining mental health outcomes in our study. This may be due to the survey timing and the respective number of reported COVID-19 cases [44].

We found that high levels of perceived health and regular physical exercise protected health workers from severe depression, anxiety, and stress. We did not find such previous findings regarding health workers. Like our findings among the general population, perceived health status was a significant predictor of depressive and anxiety symptoms [48]. Additionally, the respondents who reported no change, poor or worse physical health status, and had a psychiatric illness were significantly more likely to have higher mean DASS depression, anxiety, and stress subscale scores [49]. Similar to our findings, daily physical activity among the population was negatively associated with DASS-21 scores [50].

Our results contradict findings related to gender and offering health care services to COVID-19 patients, concerning significant factors explaining mental health outcomes [51]. This may be related to the fact that women represent nearly 70% of the global healthcare workforce [52] and 61% of the workforce in Kosovo [53]. Our sample had a more even representation of women (47%) who had participated in this survey.

## 5. Practical Implications for Health Care Policy and Management

During the pandemic, in particular, patient care requires a lot of emotional and physical energy. Mental health burden, including burnout, manifests in individuals, but it is fundamentally rooted in systems. To this end, at the highest organizational level, the accountable and measurable culture of HCW well-being shall be introduced, initially, ensuring a living wage, timely access to health services with no financial hardship, adequate sick leave [54], and replacements for those on sick and maternity leave.

Our study showed that the prevalence of stress, anxiety, and depressive symptoms among HCWs requires an immediate, specific, radical (rather than incremental) national mental health intervention program. This program would include counseling services and cognitive-behavioral training [55], which will, among other effects, help raise awareness in the early recognition of symptoms related to stress, anxiety, depression, and burnout due to workload, as well as the importance of regular physical exercise and mindfulness [56]. This program, through its project activities, would also pay special attention to the vulnerable subgroups (nurses and those working in hospitals) and address HCW-reported vulnerabilities (being married, continuously lacking an adequate supply of PPE, and feeling workplace burnout) among HCWs, such as identified in this study.

The relevant health system governance and management structures should ensure timely and continuous PPE supply through an intersectoral approach social determinants such as housing, food security [54], and childcare [57]. This mental health program should be part of the national emergency preparedness, emergency response, and health sector strategies.

The nurse workforce in Kosovo represents the majority (73.87%) of the HCWs [58]. Learning from nursing management research and practices, health organization managers should consider combining transformational and coaching leadership styles. These leadership styles represent powerful strategies to alleviate burnout by increasing mission valence. Respectively, the leaders should invest resources into developing individual team members so that external challenges are addressed while promoting personal confidence and self-awareness to work with unknown and uncertainty [59,60]. Nursing leaders should be empowered to improve work conditions and motivate nurses to decrease nurses’ burnout, reduce turnover rates, and improve the quality of nursing care [61]. Last but not least, individuals who believe they have a mission and calling to fulfill should be recruited with priority. At the same time, those already employed through continual professional education could learn to understand and live out their calling [62].

Further research in medical and occupational health should focus on establishing baseline well-being and mental health indicators (conditions/diseases and services) during recovery and normal times so they can also be used during emergencies to evaluate mental health interventions.

## 6. Conclusions

Most HCWs (71.6%) reported a mild, moderate, or severe mental health burden.

Being married, being a nurse, working in a hospital, continuously lacking adequate PPE, feeling workplace burnout, poor self-rated health, and regular physical exercise significantly predicted depression, anxiety, and stress scores.

This is the first large-sample, comprehensive, cross-sectional Kosovar study to examine the self-reported prevalence of stress, anxiety, and depression in HCW in Kosovo during the COVID-19 pandemic, followed by specific recommendations to improve the HCWs well-being, to be addressed at the health policy and healthcare management levels, during the COVID-19 pandemic and beyond.

## Figures and Tables

**Table 1 ijerph-19-16667-t001:** Characteristics of study participants.

Variable	*n*	%
Gender		
Female	256	47.0
Male	289	53.0
Age		
<60	508	94.7
>60	37	5.3
Marital status		
Single	74	13.6
Married	445	81.7
Divorced/Widow/Separate	26	4.8
Education		
High school	19	3.5
BA/Medical faculty	130	23.9
Residency	288	52.8
MA	75	13.8
PhD	33	6.1
Profession		
Medical doctor	426	78.2
Nurse	118	21.7
Working institution		
Clinic	141	25.9
Institute	29	5.3
Hospital	144	26.4
Primary care	184	33.8
Outpatient private practice	23	4.3
Administration/Other	24	4.5
Working experience		
<6 years	123	22.6
7–10 years	59	10.9
11–20 years	137	25.0
>20 years	226	41.5

**Table 2 ijerph-19-16667-t002:** Prevalence of self-reported depression, anxiety, and stress by severity.

Scores ^1^	*n*	Percentage
Depression score	545	
Normal (0–9)	432	79.3%
Mild (10–13)	37	6.8%
Moderate (14–20)	40	7.3%
Severe (21–27)	17	3.1%
Extremely severe (28–42)	19	3.5%
Anxiety score	545	
Normal (0–9)	442	81.1%
Mild (10–13)	32	5.9%
Moderate (14–20)	36	6.6%
Severe (21–27)	25	4.6%
Extremely severe (28–42)	10	1.8%
Stress score		
Normal (0–9)	371	68.1%
Mild (10–13)	55	10.1%
Moderate (14–20)	63	11.6%
Severe (21–27)	33	6.1%
Extremely severe (28–42)	23	4.1%

^1^ Scores categorized based on Lovibond and Lovibond’s percentile cutoffs (1995).

**Table 3 ijerph-19-16667-t003:** Overview of depression, anxiety, and stress levels for doctors (426) and nurses and other professional workers (118) assessed using the Depression Anxiety Stress Scales-21.

Score	M	SD	*t* Test	*p*
Depression score				
Nurses and other professional workers	5.16	4.64		
Doctors	4.19	4.65	2.0	0.04 *
Anxiety score				
Nurses and other professional workers	4.85	4.79		
Doctors	3.86	4.23	2.04	0.04 *
Stress score				
Nurses and other professional workers	5.80	4.56		
Doctors	6.02	4.94	−4.56	0.65

* *p* < 0.05.

**Table 4 ijerph-19-16667-t004:** Correlation matrix for self-reported stress, anxiety, and depression scores and their predictors.

Variable	1	2	3	4	6	7
Depression DASS-21 score	-					
Anxiety DASS-21 score	0.839 **	-				
Stress DASS-21 score	0.800 **	0.775 **	-			
Burnout	0.248 **	0.213 **	0.318 **	-		
Physical exercise	−0.162 **	−0.142 **	−0.148 **	−0.066	-	
Self-reported BMI	0.109 *	0.041	0.038	0.144 **	−0.107 *	-
Self-reported chronic illness	−0.216 **	−0.197 **	−0.129 **	−0.126 **	0.046	−0.159 **
PPE	−0.135 **	−0.146 **	−0.139 **	−0.118 **	−0.021	−0.008

* *p* < 0.05, ** *p* < 0.01.

**Table 5 ijerph-19-16667-t005:** Linear modeling analysis for predicting depression, anxiety, and stress scores (*n* = 545).

Variables	Depression				Anxiety				Stress			
Variable	B	Beta	OR	*p*	B	Beta	OR	*p*	B	Beta	OR	*p*
Gender												
Female	−0.325	0.722	0.340–1.536	0.398	−0.079	0.924	0.450–7.897	0.829	−0.280	0.757	0.355–1.612	0.470
Male	1				1				1			
Marital status												
Divorced/Widow/Separated	0.424	1.592	0.228–10.249	0.662	−0.414	0.661	0.108–4.055	0.655	−1.01	0.363	0.054–2.446	0.298
Married	−1.043	0.352	0.121–1.203	0.055	−1.194	0.303	0.110–837	0.021 *	−1.98	0.138	0.048−0.402	0.001 **
Single	1				1				1			
Profession												
Nurse	1.341	3.284	1.518–9.252	0.003 **	1.283	3.607	1.553–8.374	0.003 **	0.024	1.025	0.123–2.485	0.957
Physician	1				1				1			
Institution working												
Admin or other	0.664	1.942	0.200–18.86	0.567	1.828	6.224	0.712–54.39	0.098	1.24	3.46	0.355–33.86	0.285
Private practice	0.246	1.278	0.130–12.59	0.833	0.711	2.037	0.230–18.04	0.523	−0.245	0.783	0.079–7.555	0.834
Family medicine	1.441	4.233	0.803–22.20	0.089	1.125	3.081	0.633–14.99	0.163	0.915	2.497	0.473–13.18	0.281
Hospital	2.080	8.001	1.481–43.219	0.016 *	1.877	6.535	1.308–32.64	0.022 *	1.44	4.212	0.776–22.85	0.096
Clinic	0.977	2.656	0.488–14.45	0.258	1.182	3.262	0.649–16.41	0.151	0.201	1.223	0.224–6.682	0.816
Institute	1				1				1			
Offer health services for COVID-19 patients												
Yes	−0.259	0.771	0.231–2.572	0.771	−0.406	0.666	0.211–2.100	0.488	−0.406	0.667	0.199–2.229	0.510
No												
Available PPE at workplace	−0.497	0.608	0.409−0.904	0.014 *	−0.477	3.262	0.649–16.41	0.488	−0.401	0.669	0.450−0.996	0.048 *
Feeling workplace burnout	1.056	2.876	2.039–4.057	0.001 **	0.976	2.654	1.912–3.685	0.001 **	1.571	4.813	3.409–6.795	0.001 **
Level of self-rated health	1.361	3.899	2.734–6.403	0.001 **	1.064	3.291	1.807–4.653	0.001 **	1.392	4.022	2.466–6.613	0.001 **
Physical exercise												
Yes	−1.220	0.295	0.140−0.621	0.001 **	−1.008	0.365	0.180−0.742	0.005 **	−0.840	0.432	0.205−0.910	0.027 *
No	1				1				1			
HCW ill (COVID-19)												
Yes	−0.150	0.861	0.381–1.946	0.719	0.444	1.560	0.716–3.396	0.263	−0.299	0.742	0.327–1.681	0.474
No												
Family member ill (COVID-19)												
Yes	0.563	1.756	0.813–3.795	0.152	0.661	1.937	0.929–4.039	0.078	1.020	2.744	1.1281–6.007	0.010
No												

* *p* < 0.05, ** *p* < 0.01.

## Data Availability

Data available upon request.

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
