# Peer review of "Prevalence of Perceived Stress, Anxiety, and Depression in HCW in Kosovo during the COVID-19 Pandemic: A Cross-Sectional Survey"

_ijerph, 2022, doi:10.3390/ijerph192416667_

Round 1
Reviewer 1 Report
This submission reports the findings of a survey conducted in October 2020 in healthcare workers in Kosovo with the aim of assessing self-reported stress, anxiety, and depressive symptoms, along with their associated protective and risk factors.
In view of the paucity of data regarding how healthcare workers coped with the COVID-19 pandemic in Kosovo, this article may be suitable for publication in IJERPH. However, I sense that there is some room for improvement. Therefore, I would offer some suggestions that can hopefully help the Authors increase the value of their work. Please find my comments hereafter.
1. The last sentence of the Abstract sounds a bit vague. I would advise the Authors to rephrase it.
2. I do not get why the cutoff to stratify the population by age was set at 60 years of age, considering that it is quite obvious that healthcare workers over the age of 60 would represent a tiny minority. The Authors should justify this decision or choose a different threshold.
3. In the Discussion section (lines 245.253), I would suggest that the Authors compare their findings concerning protective and risk factors with those of a relevant meta-analysis about correlates of depressive symptoms in more than 14,000 healthcare workers amid the first phases of the COVID-19 pandemic (Crocamo et al., 2021, https://doi.org/10.1016/j.neubiorev.2021.10.010).
4. Moreover, I would suggest that the Authors take into consideration a recent and very relevant editorial published in the New England Journal of Medicine (Murthy, 2022, https://doi.org/10.1056/nejmp2207252) which offers an interesting perspective on this topic. It may provide the Authors with further cues to interpret and discuss the findings of their research.
Reviewer 2 Report
Dear authors,
I truly appreciated your efforts with mass data collection and efforts with the write-up. However, I don't see any novelty in your work. I couldn't understand the importance and contribution of the work. How do you differentiate your work from those you’ve discussed throughout the manuscript?
What was the reason for Nurses not to have a significant link with stress but with anxiety and depression? (Table 5).
- Avoid repeating sentences and outcomes in the discussion.
- How would you recommend practitioners deal with identified effects and factors for having those effects? More specifically, married participants who have a significant relationship with anxiety and stress?
Add a separate heading for theoretical and practical implications with reasonable practices and support them with references; studies with experimental and intervention programs. For more understanding, please refer to the followings.
Surviving an infectious disease outbreak: How does nurse calling influence performance during the COVID-19 fight? (2022)
Impact of nurses’ emotional labour on job stress and emotional exhaustion amid COVID-19: The role of instrumental support and coaching leadership as moderators (2022)
How to prevent and combat employee burnout and create healthier workplaces during crises and beyond (2021)
Stress, psychological distress and support in a health care organization during Covid-19: A cross-sectional study (2021)
Examining the role of transformational leadership and mission valence on burnout among hospital staff (2021)
Nurses’ Burnout: The Influence of Leader Empowering Behaviors, Work Conditions, and Demographic Traits (2017)
I hope my comments will help you to improve the content. Good luck!
Round 2
Reviewer 1 Report
The revision resulted in a much improved version of the manuscript which to me is now acceptable for publication.
Reviewer 2 Report
Dear authors,
Thanks for your efforts in revising the draft. I am truly impressed by your point-by-point response and the practical implications of the revised work. After reading the revised draft, I see significant impact of your work in the relevant and beyond fields.